# Pericentromeric heterochromatin is hierarchically organized and spatially contacts H3K9me2 islands in euchromatin

Yuh Chwen G. Lee[1]¤*, Yuki Ogiyama[2], Nuno M. C. Martins[3], Brian J. Beliveau[4], David Acevedo[1], C.-ting Wu[3], Giacomo Cavalli[2], Gary H. Karpen[1]*

1 Department of Molecular and Cell Biology, UC Berkeley and BSE Division, Lawrence Berkeley National Laboratory, Berkeley, California, United States of America, 2 Institute of Human Genetics, CNRS, Montpellier, France, 3 Department of Genetics, Blavatnik Institute, Harvard Medical School, Boston, Massachusetts, United States of America, 4 Department of Genome Sciences, University of Washington, Seattle, Washington, United States of America

¤ Current address: Department of Ecology and Evolutionary Biology, UC Irvine, Irvine, California, United States of America

* grylee@uci.edu (YCGL); gkarpen@berkeley.edu (GHK)

**Data Availability Statement:** Genome sequence (raw genome data and called transposable elements) and ChIP-seq data (raw data and

## Abstract

Membraneless pericentromeric heterochromatin (PCH) domains play vital roles in chromosome dynamics and genome stability. However, our current understanding of 3D genome organization does not include PCH domains because of technical challenges associated with repetitive sequences enriched in PCH genomic regions. We investigated the 3D architecture of *Drosophila melanogaster* PCH domains and their spatial associations with the euchromatic genome by developing a novel analysis method that incorporates genome-wide Hi-C reads originating from PCH DNA. Combined with cytogenetic analysis, we reveal a hierarchical organization of the PCH domains into distinct "territories." Strikingly, H3K9me2-enriched regions embedded in the euchromatic genome show prevalent 3D interactions with the PCH domain. These spatial contacts require H3K9me2 enrichment, are likely mediated by liquid-liquid phase separation, and may influence organismal fitness. Our findings have important implications for how PCH architecture influences the function and evolution of both repetitive heterochromatin and the gene-rich euchromatin.

## Author summary

The three dimensional (3D) organization of genomes in cell nuclei can influence a wide variety of genome functions. However, most of our understanding of this critical architecture has been limited to the gene-rich euchromatin, and largely ignores the gene-poor and repeat-rich pericentromeric heterochromatin, or PCH. PCH comprises a large part of most eukaryotic genomes, forms 3D membraneless PCH domains in nuclei, and plays a vital role in chromosome dynamics and genome stability. In this study, we developed a new method that overcomes the technical challenges imposed by the highly repetitive PCH DNA, and generated a comprehensive picture of its 3D organization. Combined

processed tracks) have been deposited to GEO (GSE125307 and GSE125031).

**Funding:** YCGL is supported by NIH K99 GM121868, YO is supported by JSPS Overseas Research Fellowships (Japan Society for the Promotion of Science), BJB is supported by Damon Runyon Cancer Research Fellowship (HHMI), CTW is supported by NIH DP1GM106412, RO1HD091797, and RO1GM123289, GC is supported by European Research Council (ERC-2008-AdG no. 232947), the CNRS, the European Union's Horizon 2020 research and innovation program under grant agreement No. 676556 (MuG), and the Agence Nationale de la Recherche (N. ANR-15-CE12-006-01), and GHK is supported by NIH R01 GM117420. The funders had no role in study design, data collection and analysis, decision to publish, or preparation of the manuscript.

**Competing interests:** Harvard University has filed patent applications on behalf of C-tW pertaining to Oligopaints and related oligo-based methods for genome imaging.

with image analyses, we reveal a hierarchical organization of the PCH domains. Surprisingly, we showed that distant euchromatic regions enriched for repressive epigenetic marks also dynamically interact with the main PCH domains. These 3D interactions are likely mediated by liquid-liquid phase separation (similar to how oil and vinegar separate in salad dressing) and the resulting liquid-like fusion events, and can influence the fitness of individuals. Our discoveries have strong implications for how seemingly "junk" DNA could impact functions in the gene-rich euchromatin.

## Introduction

Nuclear architecture and dynamics regulate many important genome functions (reviewed in [1–4]). The development of Hi-C, which combines chromosome conformation capture (3C) [5] with genome-wide sequencing [6], has led to major breakthroughs in our understanding of global nuclear architecture (reviewed in [7]). However, analyses of Hi-C results have focused on single-copy sequences in euchromatic regions (e.g., [6,8–10]), and virtually all have excluded the large Peri-Centromeric Heterochromatin (PCH) portion of genomes due to its enrichment for large blocks of repetitive DNAs [11,12]. Despite being gene-poor, the PCH plays a vital role in chromosome dynamics [13,14] and genome integrity [15–17].

A defining characteristic of heterochromatin is its enrichment for 'repressive' epigenetic features, such as Histone H3 lysine 9 di- and tri-methylation (H3K9me2/3) and its reader protein, Heterochromatin Protein 1a (HP1a) [18,19]. Interestingly, PCH DNA/chromatin from different chromosomes coalesce into one or a few membraneless PCH 'domains' (or chromocenters) in the 3D cell nucleus [20,21]. Recent studies have shown that specific biophysical properties of HP1a and liquid-liquid phase separation (LLPS) may mediate the formation of PCH domains [22,23]. This widely observed spatial organization of PCH domains could significantly influence transcription and other genome functions [24], such as silencing of euchromatic genes transposed near or in PCH genomic regions [25–27]. Furthermore, PCH-PCH interactions have recently been proposed to drive global genome architecture [28].

In addition to PCH and peritelomeric heterochromatin, regions of H3K9me2/3 enrichment are also present in the euchromatic portion of the genome [29–31]. Previous studies of a large block (~1 Mb) of *Drosophila* heterochromatin inserted in subtelomeric euchromatin ($Bw^D$) revealed that large, repetitive, H3K9me2/3 and HP1a-enriched regions in the euchromatic genome can spatially interact with the main PCH domain despite their separation by a large linear distance along the chromosome [32,33]. However, it remains unknown whether the more prevalent, smaller (tens of kbs), and naturally occurring H3K9me2/3 enriched regions in the euchromatic genome (or "H3K9me2 islands"), such as those associated with epigenetically silenced transposable elements (TEs) [34,35], also spatially contact the larger PCH domain.

We currently lack a global and in-depth understanding of the 3D organization of PCH domains, their interactions with the euchromatic genome, and the associated functional importance. To address these questions, we developed a novel method that tackles the sequence complexity of PCH to analyze Hi-C data and used it to study the 3D organization of PCH domains. Combined with cytological fluorescence in situ hybridization (FISH) analysis, we provide a comprehensive picture of the 3D structure of PCH domains in late-stage *D. melanogaster* embryos. Our analysis reveals highly heterogeneous contact frequencies among PCH regions, suggesting hierarchical ordering within the domain. Surprisingly, despite being far from PCH on linear chromosomes, euchromatic loci enriched with H3K9me2/3 can dynamically interact with the main PCH domain, and such interactions show properties consistent

with liquid-liquid phase separation and may influence individual fitness. Our study demonstrates that the spatial interactions among H3K9me2/3 enriched regions both in PCH and the euchromatic genome can have a fundamental impact on genome organization and, potentially, genome function.

## Results

### Hierarchical organizations of PCH domains

To decipher the 3D organization of PCH domains, we overcame technical limitations inherent to analyzing repeated DNA sequences and developed a new analysis method of Hi-C data that includes repetitive DNAs highly represented in PCH regions (**Fig 1A** and **S1 Fig**). The Release 6 *D. melanogaster* genome is the most complete genome among all multicellular eukaryote and includes a nearly full assembly of the non-satellite PCH DNA [36,37]. The genomic boundaries between PCH and euchromatin have also been epigenetically identified [31]. The annotated assembly allowed us to include three types of Hi-C reads that originate from PCH DNA (**Fig 1A**): 1) unique single-copy sequences within PCH (e.g., protein-coding genes, "unique"), 2) simple and complex repeats enriched in PCH ("repeat", **S1 Table**), and 3) sequences that map to multiple sites in the PCH (i.e., non single-locus mapping, "multi"). While "repeat" and "multi" categories both arise from multi-copy PCH DNA, we separated them into two categories because their identification requires different computational approaches (see Materials and Methods). We used these sequence classifications to assess contact frequencies between

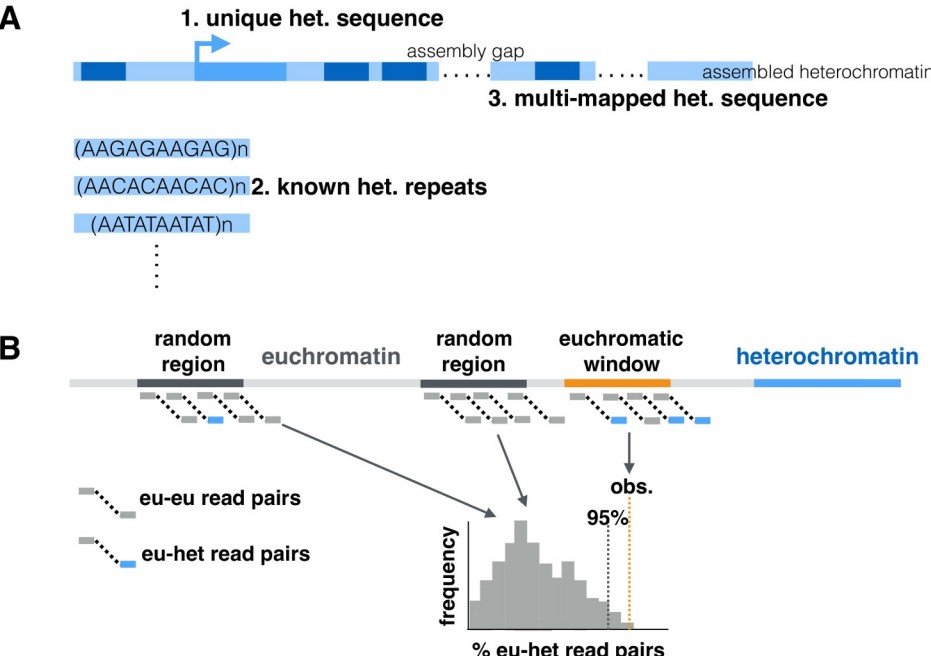

**Fig 1. New approaches for analyzing 3D organization of PCH domains.** (A) Three types of PCH-derived sequences were included in the Hi-C analysis: 1) reads mapped to single-copy sequence in the epigenetically defined PCH regions ("unique" reads, 2.4% of filtered Hi-C reads (see **S1 Fig**)), 2) reads mapped to known heterochromatic simple and complex repeats ("repeat" reads, 6.44%), or 3) reads mapped to non-unique sequences (dark blue) that are present within epigenetically defined PCH regions ("multi" reads, 3.0%). (B) Methods for assessing if an H3K9me2-enriched euchromatic region displays exceptional 3D contacts with PCH. The observed percentage of euchromatin-PCH read pairs for an H3K9me2 enriched euchromatic region is compared to a null distribution generated using randomly selected, non-H3K9me2 enriched euchromatic regions to estimate *p-value*.

PCH regions, and between PCH and H3K9me2/3-enriched regions in the euchromatic genome (**Fig 1B** and below), using published Hi-C data from 16-18hr *D. melanogaster* embryos [38].

Analyses of the formation and function of 3D PCH domains generally assume they are homogeneous, despite the fact that they contain coalesced PCH regions from different chromosomes that have high sequence heterogeneity. To investigate potential substructures within the PCH domains, we restricted the analysis to Hi-C read pairs whose both ends mapped *uniquely* to PCH genomic regions ("unique" PCH reads, **Fig 1A**) because the chromosomal locations of these reads are known. In addition to PCH regions on the 2nd, 3rd, and X chromosomes, the entire 4th and Y chromosomes were included in the analysis because enrichment for heterochromatic marks extends across both chromosomes [31,39]. We estimated the number of Hi-C read pairs coming from any two of the 100kb PCH regions. Using a sequential exclusion approach (see Materials and Methods), we identified three types of prevalent spatial interactions among PCH regions: within an arm (intra-arm), between arms of the same chromosome (inter-arm), and between arms of different chromosomes (inter-chromosome). The most frequent interactions were among PCH windows on the same chromosomal arm, which accounts for 98.08% (replicate 1, **Fig 2A**) and 97.15% (replicate 2, **S2 Fig**; and see **S3 Fig**) of parsed Hi-C read pairs (see **S2 Table** for the number of read pairs supporting each interaction). Interactions among windows within PCH arms are stronger than PCH-euchromatin interactions on the same arm (**S4 Fig**), suggesting that PCH arms (e.g., 2L PCH) are organized into distinct "territories."

Exclusion of intra-arm interactions reveals strong spatial interactions between PCH regions flanking the centromeres (inter-arm, i.e., 2L-2R, 3L-3R), which accounted for 34.72% and 35.88% (replicate 1 and 2) of the remaining read pairs (0.67% and 1.02% of total unique PCH-PCH read pairs respectively), and specific inter-chromosome interactions, mainly 3L-4 (9.68% and 9.49% of non-intra-arm read pairs). To quantitatively investigate whether these interactions are exceptional, we compared the observed percentage of read pairs against expectations that are based on either theoretical mappability [40] or empirically observed number of reads mapped to PCH on each chromosome arm (see Materials and Methods, **Fig 2B**). We also performed permutation tests for the latter to evaluate statistical significance. Contact frequencies between 2L-2R, 3L-3R, and 3L-4 are indeed significantly more than expected (compared to both expectations, permutation *p-value* < 0.0001). Finally, we excluded all intra-chromosome interactions to specifically study contact frequencies between PCH regions on different chromosomes (**Fig 2B**). The relative frequencies of most inter-chromosome associations did not exceed expectations (e.g., 2L-3L), suggesting random contacts across cell populations. However, frequencies of 3D contacts between 3rd chromosome PCH and the 4th chromosome (3L-4, 3R-4) were exceptionally high (compared to both expectations, permutation *p-value* < 0.0001). Contact frequencies between 2L-4, 2R-4, and 3R-Y were also significantly more than expected.

The spatial interactions detected with Hi-C represent a superimposition of different chromosome conformations within cell populations. To investigate the prevalence and cell-to-cell variability of identified 3D interactions, we performed single-cell FISH on embryos of the same genotype and stage as those used for Hi-C. In *D. melanogaster*, different simple repeats are specifically enriched in the PCH regions of certain chromosomes [41]. This allowed us to ask if chromosome-specific probes that label simple repeats from PCH regions that display exceptional Hi-C spatial interactions (e.g., 3R-4) colocalize more often than probes from the same chromosomes with lower frequency 3D interactions (2R-4 and 2R-3R). We measured the distance between centers of FISH signals in optical sections of 16-18hr embryos (**Fig 2C**). The distance between 3R (dodeca)-4th chromosome (AATAT) is significantly shorter than 2R

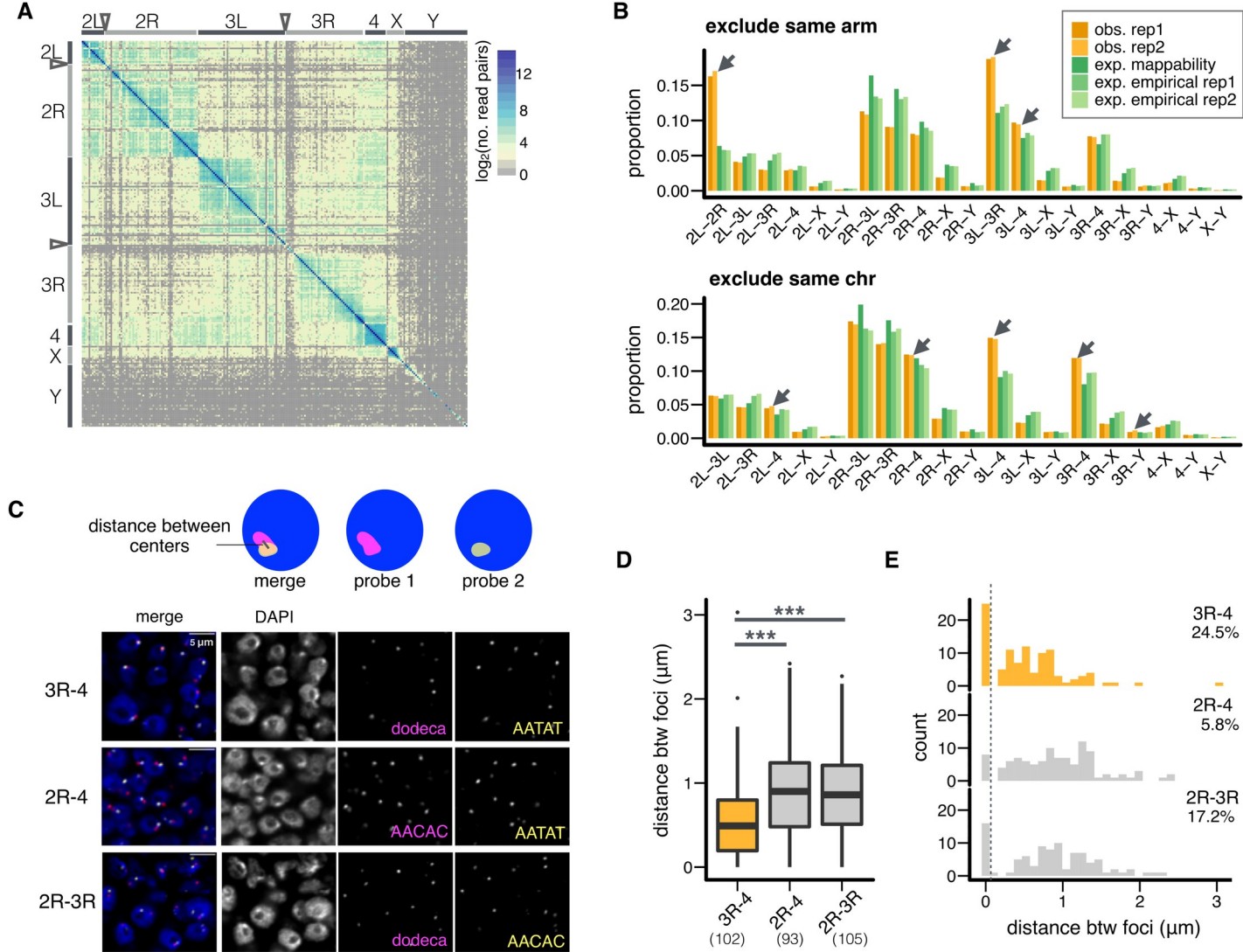

**Fig 2. Differential spatial interactions between PCH regions on different chromosomes. (A)** Heatmap for the number of Hi-C read pairs supporting the spatial interactions between pairs of 100kb PCH windows (total 189 windows). Replicate 1 is shown (see **S2 Fig** for replicate 2). Centromeres are denoted by arrowheads, and only PCH regions are shown. **(B)** Barplots for the observed and expected proportion of read pairs supporting spatial interactions between PCH on different chromosomes, excluding intra-arm (above) and inter-arm (below) interactions. Interactions that are more than expected and have significant permutation *p-values* (all $p < 0.0001$) are denoted with arrows. obs: observed proportion, exp: expected proportion **(C)** An example showing how the distance between foci was estimated (top) and representative slice images of embryonic cells stained with DAPI (DNA, blue) and FISH probes recognizing indicated PCH regions (3R-4, 2R-4, and 2R-3R, pink and yellow) (bottom). **(D, E)** Boxplot (D) and histogram (E) showing the 3D distance between PCH foci in optical sections. Orange box/bars are for exceptional PCH interactions (3R-4), while gray ones are for other interactions. In (D), the numbers of nuclei counted are in parentheses. Centerlines: median, box limits: upper and lower quartile. Points: outliers. In (E), the threshold for nuclei with overlapping foci is denoted with a dashed line, and the percentages denote nuclei with overlapping foci. *** $p < 0.001$.

(AACAC)-4 or 2R-3R (*Mann-Whitney test*, $p < 10^{-6}$ (3R-4 vs. 2R-4) and $< 10^{-4}$ (3R-4 vs. 2R-3R), **Fig 2D**). For all three pairs of interactions, the distribution of the distance between FISH signals is bimodal (**Fig 2E**), with a sharp peak near zero, and reveals a "natural threshold" dividing the nuclei into two groups (dashed line in **Fig 2E**). We defined two foci as 'overlapping' when their distances were shorter than this threshold. Consistent with the Hi-C results, the proportion of nuclei with overlapping foci was higher for 3R-4 than for 2R-4 and 2R-3R (*Fisher's Exact test*, $p = 0.001$ and $0.22$, respectively, **Fig 2E**). Overall, both Hi-C and FISH analyses demonstrate a hierarchical 3D organization of PCH domains.

## Euchromatic regions enriched for H3K9me2 show 3D contacts with PCH

The coalescence of PCH regions and large blocks of translocated heterochromatin in the euchromatic genome (e.g., $Bw^D$, [32,33]), as well as the observations of the formation of HP1a liquid droplets both *in vitro* and *in vivo* [22,23], led us to predict that small regions enriched for H3K9me2/3 and HP1a in the euchromatic genome could also spatially associate with the main PCH domains. To test this hypothesis, we identified euchromatin-PCH Hi-C read pairs, which contain sequences from single-copy, euchromatic regions paired with *any* PCH sequence (i.e., all three categories of PCH sequences, **Fig 1A**). We then estimated, among Hi-C read pairs whose one end mapped uniquely to a specific euchromatic region, the percentage of euchromatin-PCH read pairs (**Fig 1B**). We generated null distributions for the percentage of euchromatin-PCH Hi-C read pairs using randomly chosen euchromatic regions that lack H3K9me2/3 enrichment to calculate empirical *p-values* (**Fig 1B**). Euchromatic regions with exceptional percentages of euchromatin-PCH Hi-C read pairs (empirical *p-values* < 0.05) were considered to interact spatially with PCH (see Materials and Methods).

We identified by ChIP-seq 496 H3K9me2-enriched regions (defined as "H3K9me2 islands," 290bp—21.63Kb, with an average size of 3.84 kb) in the euchromatic genome (>0.5 Mb distal from the epigenetically defined euchromatin-PCH boundaries) in embryos of the same genotype and stage as the Hi-C data (see Materials and Methods). Of these H3K9me2 islands, 13.91% (n = 69) and 8.67% (n = 43) displayed significant spatial associations with PCH in either or both Hi-C replicates, respectively (**Fig 3A**). These numbers are significantly higher than expected (i.e., 5% of the H3K9me2 islands would be significant under null expectation; *binomial test*, $p = 3.04 \times 10^{-14}$ (either) and 0.00059 (both)). Thus, we conclude that H3K9me2 islands are more likely to spatially interact with PCH than euchromatic regions without H3K9me2 enrichment. For subsequent analyses, we focused on H3K9me2 islands that significantly interacted with PCH in *both* Hi-C replicates (hereafter referred to as "EU-PCH" associations).

We found that H3K9me2 islands with PCH interactions have shorter linear distance to PCH regions along the chromosome when compared to H3K9me2 islands that lacked PCH interactions (*Mann-Whitney U test*, $p < 10^{-4}$, **S5 Fig**), suggesting that proximity to PCH on a linear chromosome is a strong defining feature for the tendency to spatially interact with PCH. For each H3K9me2 island, we calculated the percentage of unique PCH reads from each chromosome arm (e.g., percentage of EU-2L PCH read pairs). For PCH region on a particular arm, H3K9me2 islands on the very same arm always have the highest such percentage (e.g., 2L euchromatic regions have the highest percentage of EU-2L PCH read pairs), followed by those on the other arm of the same chromosome (**Fig 3B** and **S6 Fig**). This echoes the observed strong tendency of "intra-arm" PCH-PCH interactions, followed by "inter-arm" PCH-PCH interactions (**Fig 2A and 2B**).

Interestingly, H3K9me2 islands that show spatial interactions with PCH have higher fractions of coding sequences when compared to H3K9me2 islands without PCH interactions (*Mann-Whitney U test*, $p = 0.0015$, median: 70.1% (with) and 30.4% (without)). In addition, these regions are more likely located within active Topologically Associated Domains (TADs) identified at the same embryonic stage [8] than H3K9me2 islands without PCH interactions (*Fisher's Exact Test*, $p = 0.0078$, **S3 Table**). Surprisingly, we also found that significant EU-PCH contacts are more likely to involve euchromatic regions in active combinatorial chromatin states [42,43]: Red or Yellow chromatin (*Fisher's Exact test*, $p = 0.021$) or modEncode States 1–4 ($p < 10^{-4}$ (S2) and = 0.011 (BG3), **S3 Table**). These regions are also depleted for chromatin states that lack obvious enrichment for histone modifications and/or protein binding: "null" TADS (*Fisher's Exact test*, $p = 0.03$), black chromatin ($p < 10^{-3}$), and modEncode

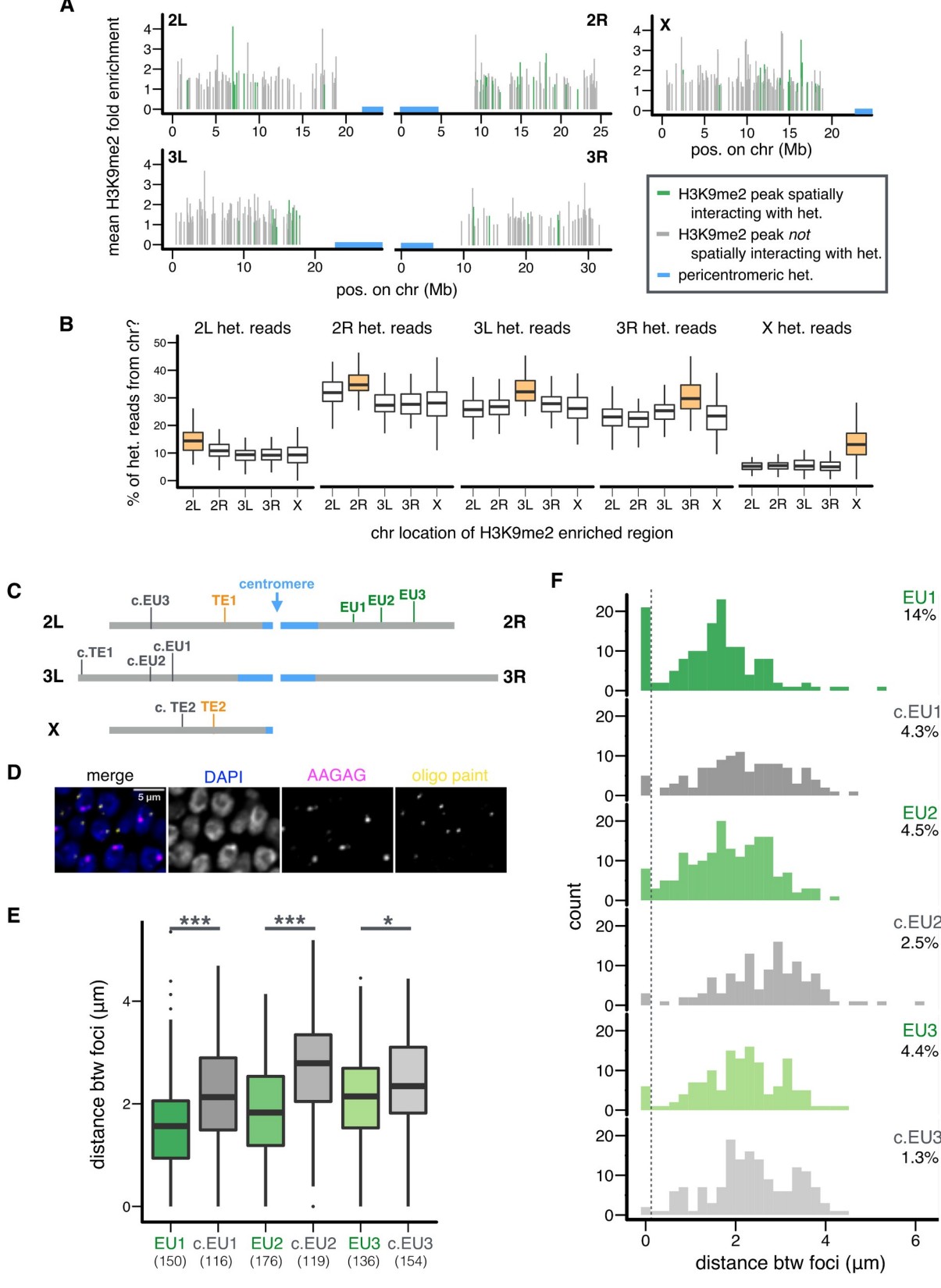

**Fig 3. H3K9me2 islands are in 3D contacts with PCH domains.** (A) Genomic distribution and average H3K9me2 enrichment level of H3K9me2 islands with (green) and without (gray) 3D interactions with PCH (blue). (B) The percentage of Hi-C reads coming from PCH regions on a particular chromosome (y-axis) is compared between H3K9me2 islands on the same (orange) or other (white) chromosomes. Replicate 1 is shown and see **S6 Fig** for replicate 2. (C) Locations of H3K9me2 islands and TEs (see below) chosen for FISH analysis. Euchromatin (gray), PCH (blue). (D) Representative slice images of nuclei stained with DAPI (DNA, blue) and FISH probes for EU1 (Oligopaint probe, yellow) and PCH (AAGAG, pink). Also see **S8 Fig**. (E, F) Boxplot (E) and histogram (F) showing the 3D distance between PCH and indicated euchromatic regions on optical sections (with PCH interaction = green, without = gray). In (E), the numbers of nuclei counted are in parentheses. In (F), the threshold for nuclei with overlapping foci is denoted with a dashed line, and the percentages denote nuclei with overlapping foci. In (B, E) Centerlines: median, box limits: upper and lower quartile. Points: outliers. $^{*}$ $p < 0.05$, $^{***}$ $p < 0.001$.

State 9 ($p$ = 0.008 (S2), **S3 Table**). Similarly, analysis using modEncode expression data from the same developmental stage (16-18hr embryos [44]) showed that genes in H3K9me2 islands with PCH interactions have higher expression than genes in H3K9me2 islands without PCH contacts (*Mann-Whitney U test*, $p$ = 0.0007). It is surprising that PCH associations would be enhanced for H3K9me2 islands containing coding genes, active chromatin marks, or higher gene expression. However, since the chromatin states and expression data analyzed are from strains that likely have different H3K9me2/3 distributions from those of the Hi-C strain, additional studies are needed. It is worth noting that PCH associations were not correlated with the following properties of H3K9me2 islands: autosome or sex chromosome linkage (*Fisher's Exact test*, $p$ = 0.27), size of the enriched region (*Mann-Whitney U test*, $p$ = 0.31), or the average level of H3K9me2 enrichment (*Mann-Whitney U test*, $p$ = 0.91). Analysis of significant EU-PCH interactions in *either* replicate reached the same conclusions (**S4 Table**).

To validate the EU-PCH 3D interactions identified by Hi-C analysis, we performed FISH using Oligopaint probes [45–47] targeting 30.5–42.9kb euchromatic regions (**S5 Table**) and probes that broadly mark PCH (AAGAG, a satellite enriched in PCH regions of all chromosomes, [48,49]). We focused on three 2R windows covering H3K9me2 islands that spatially interact with PCH (EU1-3). Because we observed that the linear distance to PCH genomic regions is a strong predictor for whether an H3K9me2 island interacts with PCH (see above), for each of these regions, we chose a matching "control" window located at a similar linear distance from PCH genomic regions that do not have H3K9me2 enrichment (c.EU1-3, see **Fig 3C** for genomic locations of chosen regions, see **S7 Fig** for their H3K9me2 enrichment level, and **Fig 3D** and **S8 Fig** for representative FISH images). Consistently, we observed that H3K9me2 islands displaying PCH interactions in the Hi-C analysis are closer to PCH in 3D space than linearly equidistant euchromatic regions that lack H3K9me2 enrichment (*Mann-Whitney U test*, $p < 10^{-5}$ (EU1 vs. c.EU1), $< 10^{-10}$ (EU2 vs. c.EU2), and 0.03 (EU3 vs. c.EU3), **Fig 3E**), confirming the Hi-C results. This difference is also reflected in the higher proportion of cells in which the two foci overlap when compared to the control regions (**Fig 3F**). It is worth noting that the comparatively lower frequency of overlapping foci for EU2 and EU3, when compared to EU1, could result from the fact that these two regions are much farther from the PCH on a linear chromosome, and thus less likely to spatially interact with PCH than EU1 (see above). This could lead to lower statistical power, and thus the comparison of proportion of overlapping foci between focused and control regions is only statistically significant for EU1 (*Fisher's Exact test*, $p$ = 0.01 (EU1 vs. c.EU1), 0.53 (EU2 vs. c.EU2), and 0.15 (EU3 vs. c.EU3)). Overall, the Hi-C and FISH analyses reveal that even short stretches of H3K9me2-enrichment in the euchromatic genome can coalescence with the main PCH domains. Note that the focused regions (EU1-3) and control regions (c.EU1-3), though similar in distance to PCH, are not on the same chromosome, and unknown biases could have led to the observed results. Stronger evidence will come from comparing the 3D organization of homologous sequences with and without H3K9me2 enrichment (see below).

## 3D PCH contacts include euchromatic TEs enriched for H3K9me2

Naturally occurring TE insertions in the euchromatic genome can acquire H3K9me2/3 marks that often extend into flanking regions, including genes [34,35,50,51], and we predict that these could also spatially contact the main PCH domains. While non-TE induced H3K9me2/3 enriched regions in the euchromatic genome are commonly *shared* between individuals (e.g., **S7 Fig**), most TE insertions are polymorphic (i.e., not present in all individuals) in the *Drosophila* population [52–54], leading to varying H3K9me2 enrichment between individuals and strains (e.g., **S9 Fig** [35]). Accordingly, to identify TE-induced H3K9me2 islands, we compared the H3K9me2 enrichment level around euchromatic TE insertions in the strain used for Hi-C (ORw1118) with that of homologous sequences in strains without the respective TEs (wild-type), as performed previously for other strains [35]. This approach identifies H3K9me2 enrichments that are broad and/or low in enrichment level, and therefore often missed by custom pipelines that rely on identifying "sharp peaks" (reviewed in [55,56]). Our analyses were restricted to 106 TEs that displayed H3K9me2 spreading into at least 1kb of flanking DNA (65% of identified TEs in strain ORw1118, see Materials and Methods), with an average of 4kb and maximum of 18kb of H3K9me2 spread. Among these TEs, 13.21% (n = 14) and 7.55% (n = 8) displayed significant spatial interactions with PCH ($p < 0.05$) in either or both Hi-C replicates respectively (see **S10 Fig** for their genomic distribution), which is significantly more than expected (*binomial test*, $p = 8.38 \times 10^{-4}$ (either) and 0.26 (both)). As a contrast, only 1.75% of TEs without H3K9me2 enrichment (n = 1) display PCH interactions. We focused on analyzing the 14 TEs showing significant PCH-contact in *either* replicate, while analyses restricted to eight TEs significant for *both* replicates were qualitatively similar (**S6 Table**). Similar to non-TE induced H3K9me2 islands, TEs spatially interacting with PCH are closer to PCH genomic regions on the linear chromosome than those that do not interact with PCH (*Mann-Whitney U test*, $p = 0.037$, **S10 Fig**). PCH-interacting TEs include those from *roo*, *pogo*, *17.6*, *mdg3*, *FB*, and *S* families. However, they were not significantly enriched for any specific TE family (*Fisher's Exact Test* for individual TE family, $p > 0.26$), class, type, or sex-chromosome linkage (**S6 Table**).

The polymorphic nature of TEs offers a rare opportunity to compare the 3D conformations of *homologous sequences* with and without TE-induced H3K9me2/3 enrichment. To validate the Hi-C results, we performed FISH analysis focusing on two TEs that are present in the Hi-C strain (ORw1118) but absent in another wildtype strain (RAL315). These two TEs also induced ORw1118-specific enrichment of H3K9me2 (**S9 Fig**) and spatially interact with PCH (TE1,2, **Fig 3C**). If the 3D proximities between the euchromatic neighborhood of these two TEs and PCH are indeed due to TE insertions instead of other properties of the regions, we would observe such spatial proximity *only* for the Hi-C strain, but not for the homologous region in the strain without the TE insertion. As controls, we also included two additional ORw1118-specific TEs that do not interact with PCH and do not have H3K9me2 enrichment (c.TE1,2, **Fig 3C**, **S9 Fig**). We predicted that the spatial distance between TE euchromatic neighborhood and PCH would *not* differ between strains *with* and *without* these two control TEs. We performed FISH using Oligopaint probes that target *unique regions* flanking the selected euchromatic TE insertions (**S5 Table**) and probes that broadly mark PCH (see **S8 Fig** for representative cell images). For TE1 and TE2, the distance to PCH signals is shorter in ORw1118 than in wildtype (*Mann-Whitney U test*, $p = 0.0008$ (TE1) and $p = 0.07$ (TE2), **Fig 4A**). Interestingly, the distribution of the distances between TE1/TE2 and PCH is bimodal for ORw1118 nuclei but unimodal for wildtype, which lacks the peaks near zero, or nuclei with overlapping foci (**Fig 4B**). Indeed, there are more nuclei with overlapping foci in ORw1118 than in the wildtype strain (*Fisher's Exact Test*, $p = 0.0007$ (TE1) and 0.070 (TE2)). Importantly, these

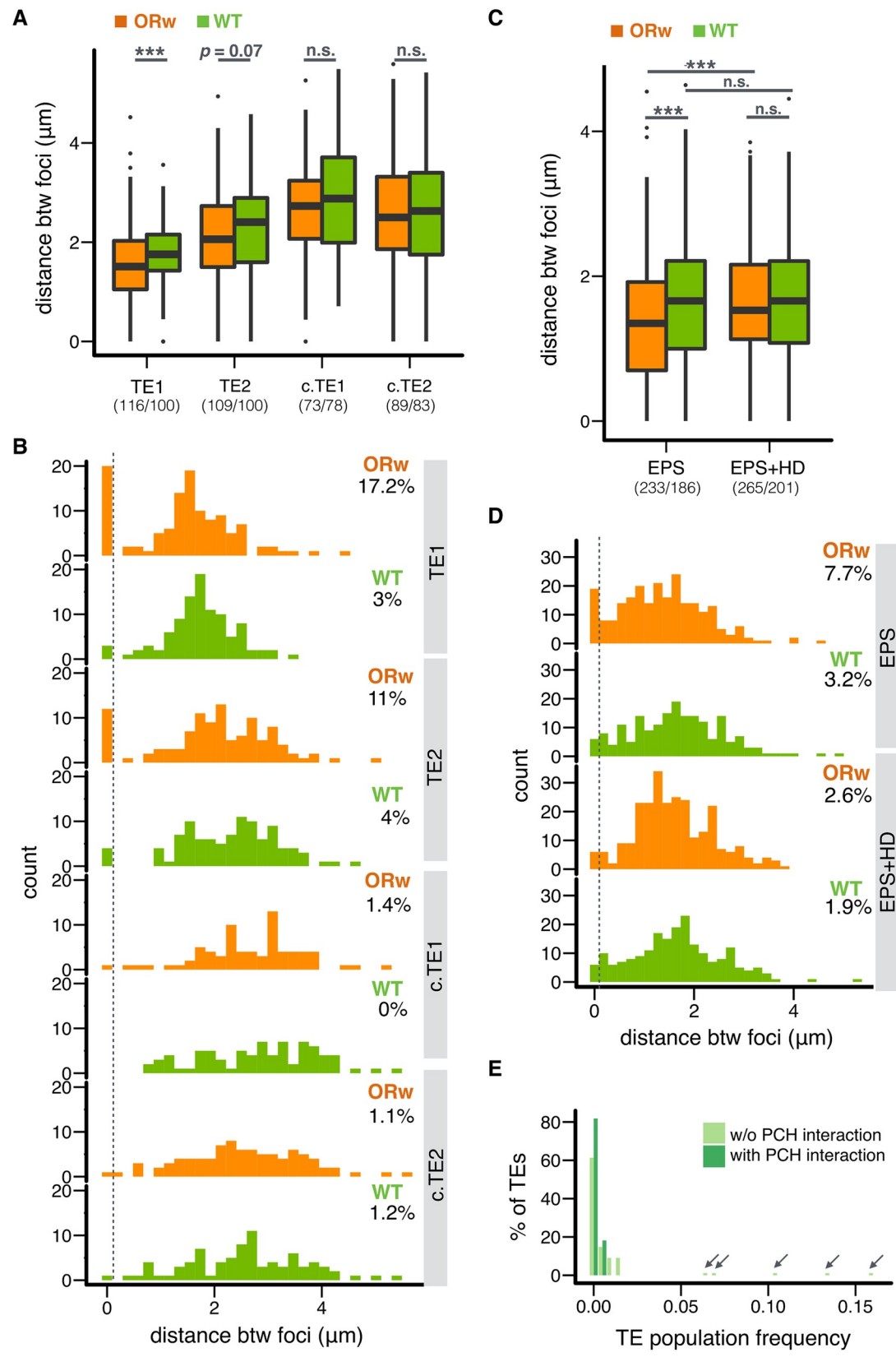

**Fig 4. Euchromatic TEs show 3D contacts with PCH, and such interactions are sensitive to 1,6-hexanediol treatment.** (A, B) Boxplot (A) and histogram (B) showing the 3D distance between euchromatic TE neighborhood and PCH on optical sections. The distance was estimated for ORw1118 (ORw, orange, TE present) and wild type (WT, green, no TE present) embryonic cells. (C, D) Boxplot (C) and histogram (D) comparing TE1-PCH distance between genotypes and between treatments. The distance was estimated for permeabilized ORw and WT embryos (EPS, see Materials and Methods) and permeabilized ORw and WT embryos with 1,6-hexanediol treatments (EPS+HD). In (A, C), the numbers of nuclei counted are in parentheses. In (B, D), the threshold for nuclei with overlapping foci is denoted with a dashed line, and the percentages denote nuclei with overlapping foci. (E) Population frequencies of TEs with and without PCH interaction. Note that high frequency TE insertions (population frequency > 0.05, arrows) all show no PCH interactions. In (A, C), Centerlines: median, box limits: upper and lower quartile. Points: outliers. *** $p < 0.001$, n.s $p > 0.05$.

between-strain differences were not observed for control TEs that lacked PCH interactions (*Mann-Whitney U test*, $p = 0.24$ (c.TE1) and 0.87 (c.TE2), *Fisher's Exact test*, $p = 0.49$ (c.TE1) and 1 (c.TE2), **Fig 4A and 4B**). This comparison of *homologous* euchromatic regions with and without TEs strongly indicates that H3K9me2 enrichment is required for spatial contacts between euchromatic regions and PCH domains.

## Euchromatin-PCH 3D contact is sensitive to perturbing liquid-liquid phase separation

The coalescence of PCH regions located on different chromosomes into 3D PCH domains in *Drosophila* exhibits properties characteristic of liquid-liquid phase separation. This includes sensitivity to 1,6-hexanediol [23], which is a mild perturbant of hydrophobic interactions [57]. To investigate if the 3D contacts between H3K9me2 islands and PCH domains is mediated by similar biophysical mechanisms, we used FISH to compare the 3D distance between PCH and H3K9me2 islands that displayed significant PCH interactions (see above) in permeabilized embryos with and without 1,6-hexanediol treatment (see Materials and Methods). We focused on TE1 because it is ORw1118-specific and leads to strain-specific H3K9me2 enrichment. This allows comparisons between genotypes with and without TEs to investigate whether the sensitivity to 1,6-hexanediol treatment is H3K9me2-enrichment dependent (see Materials and Methods, see **S11 Fig** for representative cell images). Compared to untreated controls, we observed significantly longer TE1-PCH distance (orange in **Fig 4C**, *Mann-Whitney test*, $p < 10^{-4}$) and fewer nuclei with overlapping foci (orange in **Fig 4D**, *Fisher's Exact test*, $p = 0.02$) in ORw1118 embryos treated with 1,6-hexanediol. In contrast, no such difference was observed in wildtype embryos, which do not have the TE insertion and thus no frequent TE1-PCH 3D contacts (green in **Fig 4C and 4D**, *Mann-Whitney test*, $p = 0.91$, and *Fisher's Exact test*, $p = 1$). Importantly, the significant difference in TE1-PCH 3D distance between genotypes with and without TE insertion is only observed for embryos *without* 1,6-hexanediol treatments (*Mann-Whitney test*, $p = 0.001$, *Fisher's Exact test*, $p = 0.057$), and not for those *with* the treatment (*Mann-Whitney test*, $p = 0.44$ and *Fisher's Exact test*, $p = 0.55$, **Fig 4C and 4D**). Changes in nuclear volume upon 1,6-hexanediol treatment were previously reported in cell culture [23], although we did not observe such changes in embryos with 1,6-hexanediol treatment (**S12 Fig**). Nevertheless, analysis based on the relative distance between foci (absolute distance divided by nuclear size, see Materials and Methods) gave consistent results (**S11 Fig**). The sensitivity of TE-PCH 3D contacts to 1,6-hexanediol is consistent with the spatial interactions between H3K9me2 islands and PCH domains being mediated by liquid fusions, an emergent property of liquid-liquid phase separation [23].

## Euchromatin-PCH 3D contacts may influence individual fitness

A dominant factor governing the population frequencies of TEs (presence/absence in a population) is natural selection against their deleterious fitness impacts [52,58,59]. We estimated

the population frequencies of studied TE insertions (in ORw1118 genome) in a large panmictic African population ([60], see Materials and Methods). TEs with PCH interactions have significantly lower mean population frequencies than TEs without (*t-test*, $p = 0.0042$, mean frequency $9.7 \times 10^{-4}$ (with spatial interaction) and $9.6 \times 10^{-3}$ (without), see Materials and Methods) and their frequency spectrum is more skewed towards rare variants (**Fig 4E**). Both of these observations support stronger selection against TEs with PCH interactions than other TEs [52,58,59], which could result from the negative functional consequences of TE-PCH 3D interactions. It is worth noting that even 0.01% variation in fitness, which could be rarely detected in a laboratory, can result in large differences in population frequencies in nature.

Multiple other factors have been correlated with TE population frequencies, such as TE type, chromosome linkage, and recombination rate [53,61], and could also contribute to the low population frequencies of TEs displaying PCH interactions. However, TEs with and without PCH interactions do not differ in their class, type, chromosome linkage (**S6 Table**) or local recombination rate (*Mann-Whitney U test*, $p = 0.40$). On the other hand, we observed that TEs with PCH interactions tend to be closer to genes than TEs without such interactions, although the analysis is only marginally significant (*Mann-Whitney U test*, $p = 0.065$). The stronger selection against TEs with PCH interactions could thus result from either the direct functional impact of PCH spatial contacts on adjacent genes (see Discussion) and/or other TE-mediated functional impacts along the linear chromosome (such as disrupting regulatory non-coding sequences).

## Discussion

An appreciable fraction of most eukaryotic genomes comprises constitutive heterochromatin, which is enriched for megabases of repetitive DNA localized predominantly around centromeres (PCH). However, because of technical difficulties associated with repetitive DNA, we have lacked a global and in-depth understanding of the 3D organization of the PCH domain, which encompasses at least a fifth of the human [62] and a third of the *D. melanogaster* genomes [37]. In this study, we provide a comprehensive and detailed picture of the 3D organization of PCH domains in *D. melanogaster* by combining genome-wide Hi-C analyses and cytological FISH studies. We developed a novel analysis approach that overcomes the challenges posed by repeated DNAs when determining 3D contact frequencies from Hi-C reads. Specifically, we relaxed the single-locus mapping restriction to include reads originating from the abundant repetitive DNA in PCH and used different combinations of PCH reads (single-locus mapping or not) depending on the question being addressed. Our investigations reveal significant, new insights into the interactions between different PCH regions and their 3D contacts with the euchromatic genome.

The coalescence of PCHs on different *D. melanogaster* chromosomes contributes to the formation of a large PCH domain in 3D nuclear space. However, we found that DNA contacts within the PCH domain are far from homogeneous. Our Hi-C analysis reveals the strongest interactions (~98%) involve PCH regions on the same chromosome arm (e.g., 2L), suggesting PCH regions from each arm are organized into distinct "territories" (**Fig 5**). This is similar to identified chromosome territories for the euchromatic genome [6,8,63–65]. It is clear from both the fusion of multiple PCH domains from different chromosomes [23] and our Hi-C and FISH analyses presented here that PCH regions from all the chromosomes do interact. However, some interactions occur more often than random, in particular the inter-arm (2L-2R, 3L-3R) and specific inter-chromosomal (3L/3R-4) 3D associations. Most strikingly, ~14% of identified H3K9me2-enriched regions in epigenomically defined euchromatin display preferential 3D contacts with the central PCH domains. Our quantitative FISH analysis further provides cytogenetic support for the Hi-C results. The bimodal distributions of PCH-PCH or EU-PCH

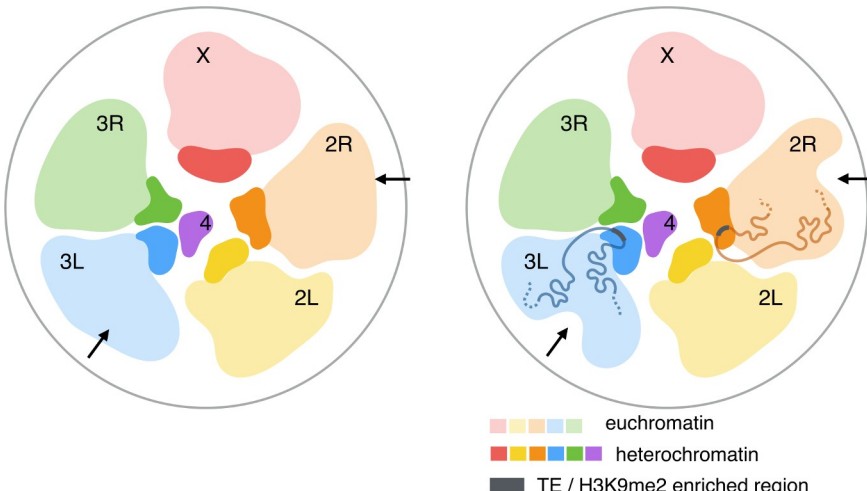

**Fig 5. Proposed spatial architecture of *D. melanogaster* genome.** PCH genomic regions located on different chromosomes coalesce to form the 3D PCH domains, or chromocenters. PCH regions (darker color) and the euchromatic genome (lighter color) form their own separate territories. PCH regions on different chromosomes interact, with inter-arm (2L-2R, 3L-3R) and inter-chromosomal 3rd-4th chromosome 3D interactions being more frequent than random expectations. 3D contacts between polymorphic H3K9me2/3-enriched regions in the euchromatic genome (potentially due to TEs, gray bar) and PCH lead to varying 3D genome conformations between individuals (arrows). 3D structures of the euchromatic genome were based on [67] and the positions of PCH regions are inferred from data in **Fig 2B**.

distances in nuclei (**Figs 2E, 3F and 4B**) demonstrate that these 3D contacts are dynamic and can vary among cells, similar to what has been previously shown for the euchromatic *Hox* loci in mouse [66]. Importantly, polymorphic TE insertions in euchromatin allowed us to directly compare homologous sequences with and without H3K9me2 enrichment, which strongly supports the conclusion that H3K9me2 enrichment is required for EU-PCH 3D contacts.

Overall, the Hi-C and FISH analyses reveal a previously unknown picture of the 3D architecture of the PCH domains (**Fig 5**): the spatial interactions within the domains, instead of being random, are hierarchical. In addition, despite the separation of euchromatic and PCH territories on the same chromosome arm [67], short stretches of H3K9me2/3 enrichment in the euchromatic genome (with and without TEs) also dynamically interact with the main PCH domains. Both PCH-PCH and EU-PCH interactions happen most often within chromosome arms, which is consistent with the predictions of polymer physics on chromosome folding [68,69]. Importantly, the tendency of H3K9me2 islands to interact with PCH strongly depends on their distance to PCH on a linear chromosome. This suggests that euchromatic regions and PCH could be in spatial proximities transiently with a frequency that largely follows the polymer physics of chromosome folding. The enrichment of H3K9me2/3 and the reader protein HP1a at specific euchromatic loci would then inevitably lead to their liquid-like fusion with HP1a-enriched PCH, resulting in frequent and/or maintained EU-PCH 3D interactions. Alternatively, this association with PCH may be an active process, regulating gene expression in specific subsets of cells. Indeed, in mice, the spatial clustering of olfactory receptor genes into heterochromatin domains silences all except for one receptor gene that spatially loops out from the cluster [70].

Our observed specific spatial contacts between PCH regions located on different chromosomes are surprising, but nevertheless consistent with the coalescence of PCH of all chromosomes into chromocenters [23]. The varying frequencies of inter-chromosomal interactions could result from non-random positioning of PCH regions upon mitotic exit [32] or constraints imposed by other nuclear structures. For example, nucleoli, whose formation is driven

by the transcription of rDNA arrays on the X chromosome, may impose structural constraints that lead to less frequent than expected spatial contacts involving X PCH. In addition, variation in biophysical properties (e.g., viscosity or varying protein compositions [71]) among PCH domains arising from specific chromosomes could result in different frequencies of liquid-liquid fusion. Indeed, the 4th chromosome has a unique composition of histone modifications and chromatin proteins [31,43] and depends on a specific suite of genes for its regulation [72,73] (e.g., requirement of Egg for histone methylation [74]), both of which could result in biophysical properties that promote frequent 3D contacts between 4th chromosome and specific PCH regions (e.g., 3L PCH).

Importantly, the population genetic analysis reveals that euchromatic TEs with PCH interactions have lower population frequencies than TEs lacking frequent PCH contacts (**Fig 4E**), suggesting that EU-PCH 3D interactions may influence individual fitness. What are the potential functional consequences of TE-PCH interactions that could influence individual fitness? TE-PCH interactions could lead to increased TE-induced enrichment of repressive epigenetic marks on neighboring sequences/genes. However, we found no difference in the extent or the magnitude of H3K9me2 spread around TEs with and without PCH interactions (*Mann-Whitney U test*, $p$ = 0.30 (extent) and 0.53 (magnitude), **S13 Fig**), suggesting that TE-PCH interactions influence other aspects of nuclear organization critical for gene regulation and/or other genome functions. For instance, 3D interactions between PCH and TEs could bring neighboring euchromatic genes into the PCH domains and result in aberrant or enhanced silencing. On the other hand, the enrichment of HP1a, and likely spatial localization in the PCH domains, can promote the expression of genes in both PCH [24,75,76] and the euchromatic genome [77–79]. Still another possibility is that the spatial contact with PCH on one chromosome may "drag" its homolog to the same nuclear compartment due to somatic homolog pairing (reviewed in [80]), resulting in *trans*-silencing [81]. A preliminary analysis found that ~15% of heterozygous TEs induced H3K9me2 enrichment not only *in cis*, but also *in trans* on the homologous chromosome without the TE insertion (i.e., *trans*-epigenetic effects, **S1 Text**). Accordingly, the fitness consequences of TE-PCH spatial interactions could potentially result from their positive or negative impacts on the expression of genes *in cis* or *in trans* to TEs, or from influencing other genome functions, such as replication and repair. Further studies are needed to test these hypotheses.

It is important to note that TEs comprise an appreciable fraction of the euchromatic genomes of virtually all eukaryotes [82]. For instance, more than 50% of assembled human euchromatin contains TEs or TE-derived sequences [62,83], many of which are interspersed with actively transcribed genes and can influence gene expression through H3K9me2/3 spreading [50]. Moreover, the presence of many TE insertions at specific locations are polymorphic between individuals in natural populations (e.g., human [84,85], *Caenorhabditis* [86,87], *Drosophila* [53,54,88], and *Arabidopsis* [89–91]). Spatial interactions between euchromatic TEs and PCH can thus generate polymorphic 3D organization of the euchromatic genomes (**Fig 5**), leading to variation in critical biological functions that depend on chromosome conformations and even varying fitness between individuals. Our investigation of the spatial architecture of PCH domains could thus have strong implications for how such 3D organizations could influence gene regulation, genome function, and even genome evolution of both heterochromatin and the gene-rich euchromatin.

## Methods

### Fly strains

Three *D. melanogaster* strains were used: Oregon-R w1118 (abbreviated ORw1118, [8]) and two wildtype strains, RAL315 (Bloomington Drosophila Stock Center (BDSC) 25181) and

RAL360 (BDSC 25186). The latter two are part of a large collection of genomically sequenced natural *D. melanogaster* strains [92], whose TE insertion positions were previously identified [88]. Flies were reared on standard medium at 25˚C with 12hr light/12hr dark cycle.

## Euchromatin-heterochromatin boundaries

To identify Hi-C reads coming from PCH genomic regions, we used epigenetically defined euchromatin-heterochromatin boundary in [31] and converted those to Release 6 coordinates using liftover (https://genome.ucsc.edu). For defining H3K9me2-enriched euchromatic regions and euchromatic TE insertions, we used 0.5 Mb inward (distal to PCH) of the epigenetically defined euchromatin-heterochromatin boundary to be conservative about what is defined as euchromatin. The entirety of 4th and Y chromosomes are enriched with heterochromatic marks [31,39] and are considered to be entirely heterochromatic.

## Generation and analysis of H3K9me2 ChIP-seq data

We performed ChIP-seq using antibody targeting H3K9me2 (Abcam 1220) on 16-18hr embryos of ORw1118 and two wildtype strains (see above). Embryo collections and ChIP-seq experiments were performed following [35], except that sequencing libraries were prepared using NEBNext Ultra DNA Library Prep Kit for Illumina (NEB cat#E7370L) following manufacturer's protocol and sequenced on Illumina Hi-Seq 4000 with 100bp paired-end reads. Each sample has two ChIP replicates (biological replicates) with matching inputs.

Raw reads were processed with trim_galore [93] to remove adaptors, low-quality bases, and single-end reads. Processed reads were mapped to release 6 *D. melanogaster* genome with bwa mem with default parameters. Reads with a mapping quality lower than 30 were removed using samtools [94]. To have enough noise for the IDR analysis (see below), we ran Macs2 [95] using broad-peak and pair-end mode, and a liberal *p-value* threshold (0.5). This was followed by performing Irreproducible Rate (IDR) analysis [96] to identify H3K9me2 enriched regions that are consistent between replicates. We defined H3K9me2-enriched regions as those with low IDR (IDR < 0.01). IDR plots for replicates for three ChIP-seq samples can be found in **S14–S16 Figs**.

## Identification and analysis of TE insertions

**TEs in wildtype strains.**   All potential TE insertions in RAL315 and RAL360 strains were previously identified using TIDAL [88]. We used the recommended coverage ratio (read number supporting TE presence/TE absence, coverage ratio at least three) to identify TEs with high confidence in these two wildtype strains. TEs in wildtype strains are used to identify ORw1118-specific TEs (see below).

**Identification of TEs in ORw1118.**   To identify TEs in the ORw1118 strain, we performed genomic sequencing. Genomic DNA was prepared from 100 ORw1118 adult female flies for each biological replicate (three biological replicates in total) with Gentra Puregene Cell kit (Qiagen cat#158388) according to the manufacturer's instructions. Whole-genome sequencing was done with overlapping 165bp pair-end Illumina sequencing on 230-240bp size genomic fragments.

We combined all three replicates of ORw1118 genomic sequencing to call TEs and quality filtered reads with Trim_galore. We identified TEs in ORw1118 also using TIDAL [88], which calls TEs with split-read methods and requires input reads to have the same length. Accordingly, we used two approaches to generate single-end reads from the original pair-end data (1) treating pair-end reads as single-end and (2) use SeqPrep (https://github.com/jstjohn/SeqPrep) to merge overlapping reads and trimmed reads to 200bp. We used the same TIDAL parameters (default) and coverage ratio (at least three) as those used in calling TEs in wildtype

strains [88]. 249 called TEs overlap between the two approaches (89.2% and 89.9% of the called TEs, respectively). We further removed TEs in shared H3K9me2-enriched euchromatic regions of wildtype strains (see above) or shared with wildtype strains, with the idea that local enrichment of H3K9me2 in ORw1118 cannot be unambiguously attributed to the presence of TE insertions. In total, 166 euchromatic TEs in ORw1118 were identified with these criteria.

To identify TE-induced local enrichment of H3K9me2, we used methods described in [35], which leverages between strain differences to identify TE-induced H3K9me2 enrichment regions with any shape, which oftentimes do not resemble peaks (e.g., **S9 Fig**). This approach is more sensitive than other custom pipelines, which look for enrichment with "peak" shape, followed by *ad hoc* merging of sharp peaks to generate "broad peak" calls (reviewed in [55,56]). We compared the enrichment of H3K9me2 in euchromatic TE neighborhoods in ORw1118 against wildtypes strains to estimate (1) the extent of TE-induced H3K9me2 enrichment (in kb) and (2) % of increase of H3K9me2 enrichment. We identified 106 ORw1118 TEs leading to at least 1kb spread of H3K9me2, with only 13 of them overlap with H3K9me2 enriched regions identified by Macs2.

We used the same approach as in [35] to estimate the population frequencies of ORw1118 TEs in an African population [60]. Similar to previously reported low population frequencies of TEs in *Drosophila* [52–54], only 36.36% of the 106 euchromatic TEs that induced H3K9me2 enrichment are present in a large African population [60] (i.e., 63.64% of those TEs are unique to ORw1118). This generally low population frequency of TEs is expected to limit the statistical power of comparison between TEs with and without PCH interactions. Indeed, we found that the median population frequencies for both TEs with and without PCH interactions are zero and not significantly different (*Mann-Whitney U test*, $p = 0.10$). Accordingly, we instead investigated whether the mean of their population frequencies differs (see main text).

### Analysis of Hi-C data

Raw Hi-C reads (two biological replicates) of 16-18hr embryos of ORw1118 from [38] were downloaded from GEO and quality filtered with trim_galore. TEs are abundant in both euchromatin and heterochromatin in *Drosophila* [37,97], and we were unable to unambiguously define which genomic compartment a TE-mapping read is from. Accordingly, we filtered reads that mapped to canonical TEs using bwa [98] and samtools [94]. Because simple and complex repeats posed serious challenges for genome assembly and are usually not included, filtered reads were then mapped to release 6 *D. melanogaster* reference genome (to identify "unique" and "multi" reads) or a list of known heterochromatic repeats (to identify "repeat" reads) using bwa with default parameters. Three types of reads are defined as from heterochromatin. (1) "unique" reads: reads that uniquely mapped (mapping quality at least 30) within epigenetically defined PCH regions in the assembled reference genome. (2) "repeat" reads: reads mapped to known heterochromatic repeats (**S1 Table**). (3) "multi" reads: reads that mapped to epigenetically defined PCH in the assembled reference genome but have mapping quality equals zero, which bwa assigns to multiple-mapped reads. Mapping locations of unique PCH reads are recorded and used for both PCH-PCH and PCH-EU analysis. The other two types of PCH reads were only used for PCH-EU analysis and their mapping locations, which are multiple in the genome, are not used. All the reads parsing were done with samtools. **S1 Fig** shows the flow chart for the filtering, mapping, and identification of PCH Hi-C reads, and the number of reads at each step. Genome-wide contact maps for both PCH and euchromatic regions (**S4 Fig**) were generated using HOMER with simple normalization [99].

**Spatial interaction between PCH regions.**   Hi-C read pairs whose both ends mapped uniquely to epigenetically defined PCH were included in the analysis. Read pairs whose mapping locations are within 10kb to each other were removed, as our analysis focuses on long-

range spatial interactions. We performed three sequential analyses (all read pairs, excluding intra-arm read pairs, excluding intra-chromosome read pairs) to identify three types of PCH-PCH interactions: within arm, within chromosome between arms (e.g., 2L-2R, 3L-3R), and between chromosomes. It is worth noting that not enough sequences have been assembled on the short arms of X, Y and 4th chromosomes, thus precluding within chromosome, between arms analysis for these chromosomes. A theoretical percentage of each pairwise interaction among PCH regions on different chromosomes was estimated based on a mappability track of *D. melanogaster* Release 6 genome, which was generated using GEM mappability tool (using read length 50 and other default parameters, [40]). We then counted the number of bases with mappability one (i.e., can be unambiguously mapped in the genome) in the PCH regions of each chromosome. Expected percentage of each pairwise interaction was also estimated empirically from the percentage of reads mapping uniquely to the PCH on each chromosome arm, ignoring read pair information. Because the Hi-C data were generated using unsexed embryos, we assumed equal sex ratio when estimating expectations. To assess whether the observed percentage is more than the empirical expectation, we randomly permuted 10,000 times read pair labels, generated an empirical distribution of the percentage, and calculated one-sided *p-values*.

**Spatial interaction between euchromatic regions and heterochromatin.**   We used samtools to parse out read pairs whose one end mapped uniquely (with mapping quality at least 30) within the focused euchromatin regions and estimated the percentage of PCH reads at the other end. All three categories of heterochromatic reads were included. Regions with less than 1,000 Hi-C read pairs were excluded from the analysis. We found strong correlations between replicates for both the percentage of euchromatin-PCH reads and the associated *p-values* (see below) for H3K9me2-enriched regions and TEs (*Spearman rank* $\rho > 88\%$, $p < 10^{-16}$, **S17 and S18** Figs). To assess whether the percentage of euchromatin-PCH read pairs is significant, we randomly selected euchromatic regions without H3K9me2 enrichment, performed the same analysis to get a null distribution of the percentage, and estimated the *p-values*. We simulated 200 sets of non-H3K9me2 enriched random euchromatic regions that are of the same sample size, on the same chromosome and, for H3K9me2 enriched regions, of the same size as the focused set. This was done separately for H3K9me2 enriched regions and TEs and separately for the two replicates. Because of the tendency of within chromosome interactions (see main text) and the difference in the PCH sizes among chromosomes [36,37], the percentage of euchromatin-PCH read pairs varies between randomly selected regions on different chromosomes (**S19 Fig**). Also, smaller euchromatic regions have fewer Hi-C read pairs included in the analysis, which translates into smaller sample size and thus larger variance of the estimated percentage (**S20 Fig**), leading to the estimates more likely to hit the boundary condition (i.e., no euchromatin-PCH read pairs, **S20 Fig,** red circles). Accordingly, for each euchromatic region, the *p-value* is estimated using random regions that are on the same chromosome and of the same size quantile. For H3K9me2-enriched euchromatic regions, we used +/-1kb of the enriched region as the defined window. Because, unlike H3K9me2 enriched regions, euchromatic TEs were identified as a small interval with possible insertions within [88], we used +/-2kb of the TE insertion site/interval as the defined window.

## Generation of FISH probes

**Heterochromatic repeat probes.**   LNA probes [100] targeting AAGAG (bulk heterochromatin), AACAC (2R PCH), dodeca (3R PCH), AATAT (4th and Y), and AATAGAC (Y) were ordered from Integrated DNA Technologies (IDT).

**Oligopaint FISH probes.**   We designed Oligopaint probes that target single copy genome regions, following [45,101]. Each targeted euchromatic region has at least 500 probes designed

to label it, with at least 12 probes/kb (**S5 Table**). For euchromatic TEs, designed Oligopaint probes target the "flanking" unique sequences instead of the TE itself. Within the total oligo library, each pool of probes targeting a genomic region was designed with an appended specific barcode (https://github.com/gnir/OligoLego, [102]), and an additional Universal barcode that were appended to the very 5' and 3' ends, both were used for PCR amplification of the specific/total library. Primary Oligopaint libraries were ordered from CustomArray (Bothell, WA), and amplified and synthesized into Oligopaint probes following [47]. To label specific subsets of oligos within the library, complementary "bridge" oligos were hybridized against their barcode, followed by hybridization with fluorophore-labeled secondary oligos complementary to an overhang of the bridge oligo. Bridge oligos and fluorescence-labeled secondary Oligopaint probe were ordered from IDT.

### Embryo collections, treatments, and fixations

**Embryo collections.**   Flies laid eggs on fresh apple juice plate for 1hr (pre-lay), followed by 2hr egg-laying on new apple juice plates. Collected embryos were incubated at 25˚C for 16hr to harvest 16-18hr embryos, which were then fixed immediately.

**Embryo permeabilization and 1,6-hexanediol treatment.**   To allow effective permeabilization of 16-18hr late stage embryos for 1,6-hexanediol treatment, 0-2hr embryos were incubated at 18˚C for 32hr, which equals to 16hr development at 25˚C [103]. Embryos were dechorionated in 50% bleach for 90s, washed with water for 1 min, and treated with EPS, a d-limonene based solvent with low toxicity [103,104], for 2 min. Permeabilized embryos were either fixed immediately or incubated in 10% 1,6-hexanediol (dissolved in PBS) for 4 min, followed by a quick wash with PBS and fixed immediately.

**Fixation of embryos.**   16-18hr embryos (without treatment) were dechorionated in 50% bleach for 90s, washed with water for 1 min. Dechorionated embryos, embryos with EPS treatment, or embryos with EPS and 1,6-hexanediol treatments were transferred to biphasic fixation solution with 4% formaldehyde (1.2mL Heptane, 75μL 16% formaldehyde, and 225 μL PBS), and shake for 20 min at room temperature. Embryos were then transferred to tubes with biphasic solution of equal volume of heptane and methanol, followed by vigorous shaking for 30-45s to crack the embryos, three washed with methanol, and stored in -20˚C in methanol.

### FISH

**Repeat probes.**   Embryos (stored in methanol) were rehydrated sequentially into PBT (1xPBS, 0.1% Tween-20), incubated with 100 μg/mL RNAseA in PBT for two hours at room temperature, washed twice with PBT, post-fixed with 4% formaldehyde in PBT for 20min, washed three times with PBT, and then sequentially transitioned into hybridization buffer (50% formamide, 5x SSC, 100 μg/mL Heparin, 100 μg/mL sheared salmon sperm DNA, and 0.1% Tween-20). Before hybridization, embryos were incubated with pre-hybridization solution (hybridization buffer boiled at 100˚C for 5 min, chilled on ice) at 56˚C for at least two hours. Embryos were then incubated with 25 ng/μL of LNA repeat probes (denatured at 70˚C for 3 min) at 80˚C for 15 min and then 37˚C with shaking overnight. For FISH with AATAT probe, embryos were incubated at 37˚C for three hours, then 25˚C overnight. Embryos were washed with hybridization buffer twice at 37/25˚C, followed by a sequential transition into PBT, two PBT washes at room temperature, DAPI staining, two PBS washes, resuspended in Prolong Gold Antifade (Life Technologies), and mounted on slides.

We used AATAT to mark 4[th] chromosome heterochromatin. Because this repeat is also abundant on the Y [41], embryos were also stained with Y-specific repeat, AATAGAC, and only female embryos were analyzed for PCH-PCH FISH analysis.

**Oligopaint probes and AAGAG probe.** Embryo FISH with both Oligopaint and AAGAG (for bulk heterochromatin) LNA probe followed [105], except for staining nuclei with DAPI and resuspension in Prolong Gold Antifade (Life Technologies).

## Imaging and data analysis

Single optical sections of 16-18hr embryos were collected on Zeiss LSM710 confocal fluorescence microscope, using a 1.4NA 63X oil objective (Zeiss), and analyzed manually in Fiji [106]. Distances between centers of FISH signals were measured using Fiji linetool in a single optical section. Only nuclei with FISH signals for both probes visible in an optical section were included in the analysis. There are usually one or two chromocenters in *Drosophila* nuclei, which is reflected in the number of AAGAG foci. For FISH using Oligopaint and AAGAG probes, when there is more than one AAGAG focus, the distance was measured between the Oligopaint focus and the nearest AAGAG focus. Because there is no difference between test and control groups in the number of AAGAG foci (% of nuclei with two AAGAG foci: ORw1118 (25%) vs. wildtype (22.9%), *Fisher's Exact test p* = 0.82; EPS treatment (14.6%) vs. EPS+1,6-hexanediol treatment (13.6%), *Fisher's Exact test p* = 1), no biases should arise due to the presence of multiple AAGAG foci. We investigated whether the nuclear volume changed upon 1,6-hexanediol treatment by measuring the radii of the DAPI region in optical sections. Although no changes in nuclear volume were observed upon 1,6-hexanediol treatment (**S12 Fig**), we also performed analysis using "relative distance" between foci, which is the absolute 3D distance divided by the radius of the DAPI region, and observed the same results (**S11 Fig**). At least 70 nuclei were counted for each treatment/genotype.

## Supporting information

**S1 Fig. Flow chart for identification of PCH Hi-C reads**
(TIFF)

**S2 Fig. Heatmap for the number of Hi-C read pairs supporting the spatial interactions between pairs of 100kb PCH windows for Hi-C replicate 2.** Note that only the PCH regions are shown.
(TIFF)

**S3 Fig. Circular plots showing inter-arm and inter-chromosomal interactions.** Circular plot showing inter-arm and inter-chromosomal interactions supported by 95, 99, and 99.9 percentile of Hi-C reads. Average mappability of each window is shown in the inner track.
(TIFF)

**S4 Fig. Genome-wide normalized contact map of replicate 1 (left) and replicate 2 (right).** Both unique euchromatic and PCH regions are shown. Blue bars are PCH regions while gray bars are euchromatic regions. Centromeres are denoted as triangles. Each element in the matrix represents the *log ratio* between the number of observed contact (Hi-C read pairs) and the number of expected contacts under the assumption that each 500kb window would have equal number of total interactions across the genome. The number of observed contacts involving Y chromosome is too low for proper normalization and thus excluded from representation in the figure. Note that this normalization may be biased against interactions involving PCH regions (EU-PCH and PCH-PCH) because much fewer reads uniquely mapped to PCH regions than euchromatic regions.
(TIFF)

**S5 Fig. Boxplot for the linear distance between H3K9me2 islands and PCH.** H3K9me2-enriched with and without PCH interactions are in green and gray respectively.
(TIFF)

**S6 Fig. Percentage of uniquely mapped heterochromatic Hi-C reads coming from a particular chromosome for euchromatic regions on different chromosomes.** Data for replicate 2 is shown.
(TIFF)

**S7 Fig. H3K9me2 enrichment level for euchromatic regions chosen for FISH analysis.** There is H3K9me2 enrichment in both ORw1118 and wildtype strains for EU1-3, but none for control regions c.EU1-3. The fourth tracks (below RAL360, blue) are broad peaks called by Macs2 in ORw1118.
(TIFF)

**S8 Fig. Representative FISH images for chosen euchromatic regions and PCH.**
(TIFF)

**S9 Fig. H3K9me2 enrichment level in euchromatic TE neighborhood.** Strain-specific H3K9me2 enrichment was observed for TE1 and TE2. Third track (one below RAL315, green) shows the insertion position of TEs in ORw1118 identified by TIDAL.
(TIFF)

**S10 Fig. Genomic distribution of TEs with and without PCH interactions. (A)** The extent of local H3K9me2 enrichment at TEs is shown on the y-axis for TEs with (green) and without (gray) local H3K9me2 enrichment, and with (dark green) and without (light green) PCH interaction. **(B)** The linear distance between PCH and TEs with (dark green) and without (light green) PCH interactions are shown in boxplots.
(TIFF)

**S11 Fig. FISH validation for the influence of 1,6-hexanediol on the spatial associations between euchromatic TE and PCH using relative distance. (A)** Representative FISH images for permeabilized embryos (EPS) and permeabilized embryos with 1,6-hexanediol treatment (EPS+HD). **(B)** Boxplot and **(C)** histogram showing the relative distance between TE1 and PCH. Comparisons of the distance between pairs of foci were tested with *Mann-Whitney test* (*p-values* in (A)) and *Fisher's exact test* (for proportion of overlapping foci, *p-values* = 0.02 (ORw, EPS vs. EPS+HD), 1 (WT, EPS vs. EPS+HD), 0.057 (ESP treatment, ORw vs. WT), 0.55 (HD treatment, ORw vs. WT)). Threshold for nuclei with overlapping foci is denoted with a dashed line.
(TIFF)

**S12 Fig. Nuclear size of embryos with and without 1,6-hexanediol treatment.** EPS: permeabilized embryos; EPS+HD: permeabilized embryos with 1,6-hexanediol treatments.
(TIFF)

**S13 Fig. Extent and magnitude of H3K9me2 enrichment of TEs with and without PCH interactions.**
(TIFF)

**S14 Fig. IDR plots for ORw1118.**
(TIF)

**S15 Fig. IDR plots for RAL315.**
(TIF)

**S16 Fig. IDR plots for RAL360.**
(TIF)

**S17 Fig. X-Y plots for the estimated proportion of euchromatin-PCH reads, and the associated *p-values*, between Hi-C replicates for H3K9me2-enriched regions.** ***$p < 0.001$.
(TIFF)

**S18 Fig. X-Y plots for the estimated proportion of euchromatin-PCH reads, and the associated *p-values*, between Hi-C replicates for euchromatic TEs.** ***$p < 0.001$.
(TIFF)

**S19 Fig. Distribution of the estimated euchromatin-PCH read pairs for random regions on different chromosomes.**
(TIFF)

**S20 Fig. Distribution of the estimated euchromatin-PCH read pairs for random regions of different size.**
(TIFF)

**S1 Table. List of heterochromatic simple and complex repeats.**
(PDF)

**S2 Table. Number of Hi-C read pairs for pairs of PCH regions.**
(PDF)

**S3 Table. Chromatin environment of euchromatic H3K9me2-enriched regions interacting with PCH.**
(PDF)

**S4 Table. Properties of euchromatic H3K9me2-enriched regions interacting with PCH.**
(PDF)

**S5 Table. Information for regions targeted by Oligopaint.**
(PDF)

**S6 Table. Properties of euchromatic TEs interacting with PCH.**
(PDF)

**S1 Text. *Trans* epigenetic effects of TEs.**
(PDF)

**S1 File. Underlying numerical data corresponding to the main and supplementary figures.**
Spreadsheet containing all raw data used to generate graphs in the main and supplementary figures.
(XLSX)

## Acknowledgments

We thank Kyle Millis for technical help with FISH, members of the Karpen lab for many helpful discussions, and Aniek Janssen, Serafin Colmenares, and Sasha Langley for carefully reading the manuscript. We thank Charles Langley for providing computational resources. We appreciate Jumana AlHaj Abed and Jeleca Erceg for helpful discussions of the Oligopaint experimental design. We also thank Vincent J. Coates Genomics Sequencing Laboratory

(GSL) at UC Berkeley, MGX sequencing and Drosophila facilities (BioCampus Montpellier, CNRS, INSERM, Université de Montpellier).

## Author Contributions

**Conceptualization:** Yuh Chwen G. Lee, Gary H. Karpen.

**Data curation:** Yuh Chwen G. Lee, Yuki Ogiyama.

**Formal analysis:** Yuh Chwen G. Lee.

**Funding acquisition:** C.-ting Wu, Giacomo Cavalli, Gary H. Karpen.

**Investigation:** Yuh Chwen G. Lee, Yuki Ogiyama, David Acevedo.

**Methodology:** Yuh Chwen G. Lee, Brian J. Beliveau.

**Resources:** C.-ting Wu, Giacomo Cavalli, Gary H. Karpen.

**Supervision:** Gary H. Karpen.

**Writing – original draft:** Yuh Chwen G. Lee.

**Writing – review & editing:** Yuh Chwen G. Lee, Yuki Ogiyama, Nuno M. C. Martins, Brian J. Beliveau, C.-ting Wu, Giacomo Cavalli, Gary H. Karpen.

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
