## [Decision Letter · Decision Letter 0]

9 Dec 2019

Dear Gary and coauthors,

Thank you very much for submitting your Research Article entitled 'Pericentromeric heterochromatin is hierarchically organized and spatially contacts H3K9me2 islands in euchromatin' to PLOS Genetics. Your manuscript was fully evaluated at the editorial level and by independent peer reviewers. The reviewers appreciated the attention to an important topic but identified some aspects of the manuscript that should be improved.

We therefore ask you to modify the manuscript according to the review recommendations before we can consider your manuscript for acceptance. Your revisions should address the specific points made by each reviewer. Because most if not all the reviewers' comments are focused on seeking clarification of methods and/or interpretation, it is likely that no new experimentation or data will be necessary and therefore your revised manuscript may not need to go back to reviewers. However, if you do choose to include new experiments/data in your resubmission, then we may need to ask reviewers for further comments.

[LINK]

Yours sincerely,

Giovanni Bosco, Ph.D.

Associate Editor

PLOS Genetics

Wendy Bickmore

Section Editor: Epigenetics

PLOS Genetics

Reviewer's Responses to Questions

**Comments to the Authors:**

Reviewer #1: The work from Gary Karpen addresses an important and not that often addressed question of nuclear biology – how pericentric heterochromatin (PCH) domain, or chromocenter, is organized in interphase nuclei. The paper presents a new bioinformatics approach, which allows to include repetitive sequences into a Hi-C analysis. Using this approach and FISH, as a confirmation of the Hi-C analysis, the authors studied 3D organization of pericentromeric heterochromatin in embryonic cells of Drosophila melanogaster.

(I) First, they used raw data from Hi-C obtained by Schuettengruber et al (2014) and found that in chromocenter, individual PCHs from different arms and chromosomes do not mix but occupy distinct territories. Moreover, they show prevalence in contacts between PCHs domains of two arms of the same chromosome, as well as specific inter-chromosomal PCH contacts. Using FISH with probes marking individual PCHs, the authors confirmed these data.

(II) Secondly, using Hi-C analysis and FISH, the authors analyzed contacts between PCH and small heterochromatic genomic foci embedded into euchromatin environment. These loci, including some transposable elements (TE), are enriched in H3K9me2 (identified in this paper by H3K9me2 ChIP-seq) and show preferential contacts between PCHs in the intra- and inter-chromosomal manner. By comparing of TE with and without enrichment with H3K9me2, the author show that this histone modification is essential for attraction of foci to PCH.

(III) Furthermore, the authors suggest, that coalescence of PCH regions and foci is mediated by accumulation of HP1, chromatin binding protein with known properties to form liquid separate phase. They show, that after disruption of liquid-liquid phase separation, the genomic foci lose their attraction to PCH domain.

(IV) And finally, the authors estimated the frequency of TE insertion in a large population of D.melanogastra using dataset published by Lack et al (2015) and show that TE identified in this study as interacting with PCH are more rare that TE without such interactions. Based on this observation, the author speculate that association of loci with PCH may influence individual fitness.

All in all, I think this study is interesting and important for the nuclear biology and more generally to cell biology. Since I cannot judge the bioinformatics aspect of the work, I will comment only on microscopy.

I have two main criticism to the microscopic experiments:

(1a) First of all, I do not understand how the authors measured the distances between FISH signals. In the figure 2C, they indicate “distances between centroids”. In the M&M section they write that “Distances between foci were measured by Fiji linetool”. The latter excludes the former. Linetool can be used either on projections or on single optical sections. Since the authors claim that they performed 3D analysis of the PCH and other chromosomal regions, I expect that 3D distances are calculated from measured geometrical centers (e.g., using Fiji). Was it the case?

(1b) A related question: how the authors estimate “overlapping”? Was it actually measured or estimated by eye? And what is the “natural threshold”? (lines 19-22). I believe, the authors have to explain how the distance measurements and scoring was performed in more detail.

(2) Secondly, I do not understand why the authors prefer relative distances to absolute distances. Interactions between foci, especially in case of liquid-liquid phase separation, are physical interactions, they either exist or do not exist, and whether they occur in a small or large nucleus is not important. Besides, as I can judge from images in the figures, the differences in the nuclear sizes are pretty small. Moreover, it seems to me that nuclear radii were defined arbitrary, probably using the Fiji linetool. For proper measurements of a nuclear radius, one would need to segment a nucleus and define its geometrical center, on a first place. Although in M&M part (lines 11-18) there is no indication of how the radii ware measured, I do not think the authors performed such measurements.

Minor comments:

(3a) All microscopic figures: According to a good publishing practice, all single channel panels have to be presented as grey scale images, while leaving RGB panels only for overlays of false colored channels.

(3a) All microscopic figures: The authors should indicate whether presented images are single confocal sections or projections. If projections, they should indicate of how many sections (distances).

(3a) All microscopic figures have no scale bars.

(4) Figure 4: Why there is such a big difference between TE1-PCH overlapping frequency on graph 4B and 4D, 17.2% and 7.7%, respectively?

(5) On some of DAPI nuclear images (when converted to grey scale), one can see not a single chromocenter but two or at least a bipartite structures intensely stained with DAPI. I am not familiar with Drosophila nuclei, but could it be that a chromocenter is split in two? How has this influenced measurements? The authors have to mention this point.

(6a) The authors do not discuss what causes the hierarchy in the chromocenter structure with PCH. It seems that PCH on X chromosomes is relatively distanced from other PCHs. Could it be because the X-chromosome in Drosophila harbors NOR and the formed nucleolus might impose some geometrical constrains?

(6b) Can one explain the different degree of clustering or chromosomes 2 and 3 with entirely heterochromatic chr.4 by different amount of PCH on these chromosomes?

(7) I am missing a discussion about the phenomenon of contacts between “H3K9me2 islands” and PCH. Do authors explain such contacts as a mechanism of genome regulation or simply as an inevitable consequence of liquid-liquid phase separation caused by HP1 enrichment in both loci? If it is important for transcription regulation, is similar phenomena exist in nuclei with polytene chromosomes?

(8) The authors need to specify which age of D.melanogaster embryos was used by Schuettengruber et al (2014) – in their paper I found only indication that ChIP experiments on whole Drosophila embryos were performed 4–12 hr after egg laying, which is different from the age of embryos used by Lee at al (16-18 hr)

(9) In M&M, there is no information about how confocal stacks were acquired (voxel size, lasers, etc) and whether axial chromatic shift correction was performed

(10) Number of Supplementary figures is unnecessarily high and should be reduced, for instance, figures S4 and S5 can be joined; the same is for figures S13 and S14.

(11) The left part of the schematics in Figure 5, in my view, can be omitted

Reviewer #2: The review is uploaded as an attachment.

Reviewer #3: In the manuscript " Pericentromeric heterochromatin is hierarchically organized and spatially contacts H3K9me2 islands in euchromatin ", Lee et al present a study of the 3D structure of pericentromeric heterochromatin (PCH) domain in Drosophila melanogaster embryo. They developed a new approach to infer information about the contacts between the PCH domains, and also between PCH domain and euchromatin using Hi-C reads. Using this approach, they discovered that PCH domains are organized hierarchically inside nuclei, with strongest contact of PCH domains within the same chromosome arm, and then contact of PCH domains from different arms but in the same chromosome (e.g. 2L-2R), they also discovered specific inter-chromosomal interactions (e.g. 3L-4). Interestingly, they identified that euchromatic H3K9me2/3 islands, including those presumably induced by TE insertions, interact with PCH domains. Their finding that low frequency TEs have more PCH interactions suggests that TEs that cause interactions between euchromatin and the PCH are more deleterious.

This is an exciting paper with important implications for genome organization and evolution. Overall, the rationale for their study is well defined, and their conclusions about PCH hierarchical organization is well supported. However, I think that some of their analyses require additional justification.

I have outlined my concerns below:

Major comments:

-The justification for the “unique”, “repeat”, and “multi” categories was unclear to me. Why did the authors separate reads from multicopy DNA into “repeat” reads and “multi” reads into two categories? How did this partitioning affect their results?

- Do the euchromatic H3K9me2/3 islands interact with other euchromatic H3K9me2/3 islands on the same chromosome? If all of the islands interact with their own PCH, then you would expect this to be the case, but these interactions may be hierarchical even within an arm. If so, intra-arm Eu H3K9me2/3 interactions (compared to inter-arm Eu H3K9me2/3 interactions) might be stronger and easier to see than PCH-Eu interactions.

-The authors showed that euchromatin-PCH interaction is sensitive to perturbing liquid-liquid phase separation. Did they confirm that the PCH-PCH domain interactions are also perturbed in their experiments?

-The motivation for the experiments described on page 18 and in figure 4 was not clear. The result is interesting, but may require reframing. The authors say that polymorphic TEs allowed them to determine the effect of TE-induced H3K9me on euchromatin-PCH interactions. However, the control TEs (c.TE1 and c.TE2) are those without H3K9me2/3 enrichment and without interaction with PCH, but they are also much further away from the PCH on the linear chromosome than the polymorphic TEs (TE1 and TE2) that interact with the PCH and have H3K9me enrichment. It seems that all that can be concluded from this experiment is that a transposable element alone is not sufficient to cause an interaction with the PCH. Perhaps a better control for asking about the effect of TE-induced H3K9me2/3 on interaction with PCH is to compare these TEs with a pair equally close to the PCH but without H3K9 methylation.

Minor:

Page 24 line 20: “PHC” should be “PCH”

Page 37 lines 12, 15, 18: fix ug and uL character

Fig 3F: threshold for overlapping might work better as a dotted line through each plot than an arrow on the first plot

**Have all data underlying the figures and results presented in the manuscript been provided?**

Reviewer #1: Yes

Reviewer #2: Yes

Reviewer #3: Yes

PLOS authors have the option to publish the peer review history of their article (what does this mean?). If published, this will include your full peer review and any attached files.

Reviewer #1: No

Reviewer #2: No

Reviewer #3: No

---

## [Editor Report · Decision Letter 1]

14 Feb 2020

Dear Gary and colleagues,

We are pleased to inform you that your manuscript entitled "Pericentromeric heterochromatin is hierarchically organized and spatially contacts H3K9me2 islands in euchromatin" has been editorially accepted for publication in PLOS Genetics. Congratulations!

Yours sincerely,

Giovanni Bosco, Ph.D.

Associate Editor

PLOS Genetics

Wendy Bickmore

Section Editor: Epigenetics

PLOS Genetics

Comments from the reviewers (if applicable):

**Data Deposition**

http://datadryad.org/submit?journalID=pgenetics&manu=PGENETICS-D-19-01791R1

**Press Queries**

---

## [Editor Report · Acceptance letter]

13 Mar 2020

PGENETICS-D-19-01791R1 

Pericentromeric heterochromatin is hierarchically organized and spatially contacts H3K9me2 islands in euchromatin 

Dear Dr Lee, 

We are pleased to inform you that your manuscript entitled "Pericentromeric heterochromatin is hierarchically organized and spatially contacts H3K9me2 islands in euchromatin" has been formally accepted for publication in PLOS Genetics! Your manuscript is now with our production department and you will be notified of the publication date in due course.

With kind regards,

Kaitlin Butler

PLOS Genetics

On behalf of:
